# Sexually Transmitted Infections and Behavioral Determinants of Sexual and Reproductive Health in the Allahabad District (India) Based on Data from the ChlamIndia Study

**DOI:** 10.3390/microorganisms7110557

**Published:** 2019-11-12

**Authors:** Pierre P. M. Thomas, Jay Yadav, Rajiv Kant, Elena Ambrosino, Smita Srivastava, Gurpreet Batra, Arvind Dayal, Nidhi Masih, Akash Pandey, Saurav Saha, Roel Heijmans, Jonathan A. Lal, Servaas A. Morré

**Affiliations:** 1Institute of Public Health Genomics, Genetics and Cell Biology Cluster, GROW Research School for Oncology and Development Biology, Maastricht University, 6229 ER Maastricht, The Netherlands; e.ambrosino@maastrichtuniversity.nl (E.A.); jonathanalal@shiats.edu.in (J.A.L.); 2Department of Molecular and Cellular Engineering, Jacob Institute of Biotechnology and Bioengineering, Sam Higginbottom University of Agriculture, Technology and Sciences, Allahabad, Uttar Pradesh 211007, India; biotech.jai@gmail.com (J.Y.); rajiv.kant@shiats.edu.in (R.K.); masih.nidhi@gmail.com (N.M.); akashpandey971997@gmail.com (A.P.); 3Hayes Memorial Mission Hospital, Shalom Institute of Health and Allied Sciences, SHUATS Allahabad, Uttar Pradesh 211007, India; Drsmitasrivastava75@gmail.com (S.S.); gurbatra@gmail.com (G.B.); arvindayal@gmail.com (A.D.); 4Department of Computational Biology and Bioinformatics, Jacob Institute of Biotechnology and Bioengineering, Sam Higginbottom University of Agriculture, Technology and Sciences, Allahabad, Uttar Pradesh 211007, India; saurav.saha@shiats.edu.in; 5Laboratory of Immunogenetics, Department of Medical Microbiology and Infection Control, VU Medical Center, 1081 HV Amsterdam, The Netherlands; r.heijmans@vumc.nl

**Keywords:** *Chlamydia trachomatis*, *Neisseria gonorrhoeae*, India, sexual and reproductive health

## Abstract

Background: Sexually transmitted infections (STIs), like *Chlamydia trachomatis* and *Neisseria gonorrhoeae* (CT and NG, respectively) are linked to an important sexual and reproductive health (SRH) burden worldwide. Behavior is an important predictor for SRH, as it dictates the risk for STIs. Assessing the behavior of a population helps to assess its risk profile. Methods: Study participants were recruited at a gynecology outpatient department (OPD) in the Allahabad district in Uttar Pradesh India, and a questionnaire was used to assess demographics, SRH, and obstetric history. Patients provided three samples (urine, vaginal swab, and whole blood). These samples were used to identify CT and NG using PCR/NAAT and CT IgG ELISA. Results: A total of 296 women were included for testing; mean age was 29 years. No positive cases of CT and NG were observed using PCR/NAAT. A 7% (22/296) positivity rate for CT was observed using IgG ELISA. No positive association was found between serology and symptoms (vaginal discharge, abdominal pain, dysuria, and dyspareunia) or adverse pregnancy outcomes (miscarriage and stillbirth). Positive relations with CT could be observed with consumption of alcohol, illiteracy, and tenesmus (*p*-value 0.02–0.03). Discussion: STI prevalence in this study was low, but a high burden of SRH morbidity was observed, with a high symptomatic load. High rates of miscarriage (31%) and stillbirth (8%) were also observed among study subjects. No associations could be found between these ailments and CT infection. These rates are high even for low- and middle-income country standards. Conclusion: This study puts forward high rates of SRH morbidity, and instances of adverse reproductive health outcomes are highlighted in this study, although no associations with CT infection could be found. This warrants more investigation into the causes leading to these complaints in the Indian scenario and potential biases to NAAT testing, such as consumption of over-the-counter antimicrobials.

## 1. Introduction

Sexual and reproductive health (SRH) is an essential component of human well-being [1]. It is in fact important to ensure that every member of society benefits from healthy sexual interactions and is given the opportunity to live their sexuality and to start a family. Improving sexual and reproductive health contributes to healthy and fulfilled populations [2]. Some of the most important threats to this well-being are sexually transmitted infections (STIs). These can be passed on between a partner that is infected and another who is exposed and susceptible to these infections. STIs can cause a vast array of consequences, both in the short and long term [3]. Acute complaints include vaginal and urethral discharge, lower abdominal pain dysuria, and dyspareunia [4]. Untreated infections may lead to chronic impairments that comprise infertility, chronic pelvic pain, ectopic pregnancy, pelvic inflammatory disease (PID), and tubal factor infertility (TFI). These concerns hold particular importance for low- and middle-income countries [5], since they suffer greatly from reproductive morbidity. Infectious diseases, including STIs, are also endemic there, and are spreading among the communities. Tracking the disease trends and epidemiology of these diseases is of particular importance, as many STIs can remain undiagnosed and untreated due to their asymptomatic nature [6]. Acting to prevent the spread of STIs in those populations is hence a matter of public health importance [7]. STIs can be addressed effectively through treatment once they are diagnosed properly.

Behavior is an important component in SRH [8]. Behavioral and demographic determinants in fact play an important role in the spread of STIs, as well as in the development of long-term health consequences [9]. Certain behaviors may put individuals at a higher risk for the acquisition of STIs, such as by *Chlamydia trachomatis* (hereinafter CT) and *Neisseria gonorrhoeae* (hereinafter NG). These diseases are among the most prevalent worldwide, with over 300 million prevalent cases of CT in the world [10]. These diseases are closely associated with acute clinical presentations and lifetime impairments. Some exposures, such as commercial sex work and transactional sex, may serve as an indicator for an individual’s risk profile [11,12]. Certain practices have furthermore been proven to contribute to the spread of STIs in the community [13].

Furthermore, the SRH history, more specifically the obstetrical and gynecological history, can serve as a marker of past exposure to bacteria like *C. trachomatis* and *N. gonorrhoeae*, and to other pathogens like the human papilloma virus (HPV) and the human immunodeficiency virus (HIV/AIDS) [14,15]. Sexual and reproductive health is furthermore a contentious topic in India, as sexual activity is shrouded in taboo and stigma both against patients with HIV and STIs, as well as against infertile patients [16,17]. This also holds true for the state of Uttar Pradesh and the district of Allahabad. These localities are in fact home to high population densities, high poverty, and low development [18]. This warrants investigation of the CT burden, as vulnerable groups might be at a high risk to develop adverse and debilitating health consequences.

Understanding the spread of STIs and the role of certain behaviors in the transmission of sexually transmitted pathogens constitutes a stepping stone towards better prevention strategies [19,20]. Indeed, understanding the risk behaviors that put the population at risk of getting infected with *C. trachomatis* and other infections has already helped to tailor prevention efforts, and assist in the management of these ailments in the past [21].

This paper presents the findings of the ChlamIndia project conducted in the district of Allahabad, in the northern Indian state of Uttar Pradesh. This paper compiles the findings of the study on the topics of SRH, STIs, and behavioral outcomes in a group of women attending the OPD at a local hospital for testing and detection of *C. trachomatis* and *N. gonorrhoeae*.

## 2. Materials and Methods

### 2.1. Patient Population

The prospective recruitment of the study participants took place between 2016 and 2018 at the OPD of the Hayes Memorial mission hospital located in the Naini locality of the Allahabad district in the state of Uttar Pradesh, India. Participants were included among the attending population at the OPD. Potential study participants were screened by the referring gynecologists based on medical reproductive history and symptoms. Informed consent for the participation was taken by the attending physician from the selected participants. The study and its protocol were approved by the Institutional Ethical Committee for Biomedical Research on Human Participants at the Sam Higginbottom University of Agriculture Technology and Science (Reference number: IEC/SHUATS/2017/B/01, approved 30 April 2016).

### 2.2. Clinical Samples and Diagnostic Tests

After inclusion in the study, the patients were asked to provide three samples: a vaginal swab collected using COPAN FLOQSwab and preserved in eNAT medium (Copan diagnostics, Brescia, Italy) and 50 mL of first void urine (FVU) destined for testing using nucleic acid amplification techniques (NAAT) (Roche LightCycler 480, PRESTO 500 CT–NG testing kits, Goffin Molecular Technologies, Beesd, The Netherlands). Twenty milliliters of the urine samples were centrifuged into pellets; those were concentrated in 1 mL of substrate and processed according to the high pure PCR template preparation (HPPTP) extraction process. The DNA was extracted and prepared for PCR analysis (LED Cycler, Roche, Basel, Switzerland) using the Roche HPPTP kits (Roche Diagnostics, Basel, Switzerland). Thirdly, a 20 mL blood sample was collected for IgG ELISA serology analysis. The samples were centrifuged and separated into plasma, and peripheral blood mononuclear cells (PBMCs) were extracted. Plasma was prepared for ELISA serology testing using the Medac CT IgG pELISA PLUS (Medac Diagnostics GMBH, Weidel, Germany) according to the instructions of the manufacturer (Chlamydia plus IgG test, Medac GMBH, Weidel, Germany). Positive controls were found to be positive and the IACs (isolation and amplification controls) were positive in all samples, meaning the DNA isolation and the PCR were performed correctly and efficiently.

### 2.3. Qualitative Data

Patients were administered a questionnaire by the attending physician in English, or in the local language (Hindi). The same person also filled in the questionnaire. The questionnaire was split into several parts, namely: demographics, symptoms and presentations, sexual and reproductive health history, obstetrical and gynecological history, and exposure to risk behavior. The questionnaire consisted of a total of 177 questions. A panel of questions were selected for analysis based on their relevance for the scope of this study.

### 2.4. Data Analysis

Data from the questionnaires were included in an anonymized database for research use. Ten percent of the content of the database was cross-checked for integrity using the hard copy of the questionnaire filled in by the physician. Data was entered on a password-protected excel sheet. Data analysis was performed on an IBM SPSS Statistics Processor 24.

## 3. Results

### 3.1. Demographics

The questionnaire data were obtained for a total of 296 participants. Details of the study population are summarized in Table 1 below.

The study population was characterized by a large age range (18–72), with most of the participants being housewives. The most prevalent population subgroup were Hindus, followed by Muslims and Christians.

### 3.2. Reproductive Health Morbidity and Obstetric History

PCR analysis of both vaginal swabs and urine samples did not yield any CT or NG positive results. ELISA CT IgG serology highlighted the presence of IgG antibodies in 22 out of 296 patients (7.4%).

### 3.3. Symptoms and Presentations

Almost all patients who tested positive were symptomatic. In fact, discharge was observed in 21 out of 22 patients who tested positive on the ELISA. The symptoms and relevant SRH presentations within the testing population are featured in Table 2 below:

A pattern of high symptomatic load was observed in the studied population, with most patients reporting complaints such as abdominal pain, discharge, and dysuria. This high symptomatic load did not yield any significant relationships with chlamydial or gonococcal infections. These symptoms were in fact not significantly associated with positive serology outcomes for CT, as highlighted by the *p*-values (obtained through Fisher’s exact test). A significant association could, however, be observed between CT seropositivity and tenesmus, as well as with the regularity or irregularity of the menstrual pattern.

### 3.4. Behavioral Outcomes

Although most of the study population (94%, 281/296) reported being sexually active in the last six months before testing, behaviors that could be interpreted as promiscuous were very scarcely recounted. Indeed, there were no instances of sex with multiple partners among the participants. Oral and anal sexual acts were also very rare. There was in fact one woman (0.34%) reporting sometimes engaging in oral sex, while the rest of study participants stated that they never did it. Anal sexual contact was similarly only conveyed by one woman (0.34%), with the other women having never been involved in such practice (99.6%, 295/296). A summary of relevant behavioral practices is provided in Table 3 below:

Significant associations (using Fisher’s exact test) could be observed between the consumption of alcohol and positive serology results for CT. Having discussed any clinical complaints with one’s partner was also associated with *C. trachomatis* infection, although this association was not statistically significant. There was also a suggestive relation between early sexual debut (between 13 and 20 years of age) and a positive serology result for *C. trachomatis*, although this relation could not reach significance either.

### 3.5. History of Adverse Pregnancy Outcomes

Obstetrical and gynecological history, as well as the amount of adverse pregnancy outcomes (miscarriages, stillbirths), and their relation to *C. trachomatis* seropositivity, are summarized in Table 4 below:

Although none of the obstetrical and gynecological outcomes reached statistical significance, the proportions in which they presented in the tested population was high. This is in fact an indication of a high reproductive health burden in the studied community. Furthermore, some of the trends might have reached statistical significance with a larger sample size.

## 4. Discussion

This paper compiles the findings of the ChlamIndia study that utilized NAAT and serology to identify the presence of *C. trachomatis* and *N. gonorrhoeae* in a semi-rural cohort of women from the Allahabad district. No CT or NG infections could be detected by PCR/NAAT testing. Serological analysis of the samples indicated that CT positivity was linked, or closely linked (*p* = 0.02–0.07), to tenesmus, irregular menstruation, and discussing complaints with one’s partner.

It should be clear that CT positivity in our study was based on CT IgG positivity, rather than CT DNA positivity. The fact that all samples were negative for CT and NG DNA was quite unexpected and led to the consideration of the following steps: (1) The PCR was performed by a local technician in India, trained by an experienced Dutch researcher who also controlled all of the results. No technical issue suggestive of potentially false negative results was encountered at this stage; (2) The equipment used in India was identical to, and validated and used in the same way as the equipment used in The Netherlands, suggesting no equipment-based issues [22]; (3) Serial dilutions of positive controls gave the same range in both India and The Netherlands, suggestive of no sensitivity issues at the Indian test site; (4) All PRESTO assay IACs were positive, indicating good DNA isolation and efficient PCR performance without inhibition; and (5) A series of samples were tested for a human HLA target to ensure the samples contained human DNA. All were positive, suggesting correct sample collection. Furthermore, the PCR/NAAT assay used in this study had previously been employed in similar studies that were conducted in, and on samples from, both high and low-resource settings (The Netherlands, South Africa, and Tanzania) [22,23]. These studies were successful in identifying CT and NG DNA while making use of the same techniques. This highlights that the encountered CT and NG DNA negativity cannot be attributed to the testing methodology. Several explanations for these results were considered. A possible explanation is the widespread use of over-the-counter (OTC) antibiotics in the Indian setting. Antibiotic drugs can in fact be obtained easily in both urban and rural settings, and may be dispensed by pharmacists rather than physicians based on reported symptoms and presentations rather than on testing results [24,25]. These practices are amongst the most important drivers of drug resistance in the Indian setting [26]. Such a hypothesis is in line with the very common irrational use of antibiotics for SRH complaints reported by the gynecologists and physicians involved in the study (personal communications). Common drugs consumed over the counter in the community included Metronidazole, Ofloxacin, as well as 2nd and 3rd generation cephalosporins (based on personal communications with the physicians). All of the commonly used (and possibly misused) drugs by the study population have an effect on chlamydial infections, and may hence have influenced the results of PCR testing [27,28]. This makes the conclusion that potential misuse of antimicrobial drugs in the study population a plausible explanation.

It is in fact common for patients and women to delay consultation with a gynecologist until symptoms and complaints can no longer be managed with empirical, over-the-counter drug regimens. The fact that non-specific STI-related complaints (vaginal discharge, abdominal pains, and dysuria) were recorded in a non-negligible subset of the population, and the observation of high numbers of adverse pregnancy outcomes among some study participants, suggest the presence of a reproductive health burden, which could be attributed to, amongst others, sexually-transmitted pathogens. It is nonetheless agreed upon that common STI presentations like vaginal discharge, abdominal pain, dysuria, and dyspareunia are not specific, and that even the most recent algorithms for syndromic management do not allow for precise diagnosis merely based on symptoms [29]. This is particularly true for vaginal discharge, as it can present physiologically or pathologically in women [30]. These presentations may in fact be associated with a wide range of pathogens and/or conditions. A study by Brabin et al. reported similarly low levels of PCR-positivity in a diverse cohort of mostly symptomatic women (typically vaginal discharge) in Mumbai. The study, however, identified high rates of different causative agents, such as *Ureaplasma urelitycum* and *Mycoplasma hominis* [31]. It is possible that the symptoms in the present study may have been linked to infections with other pathogens. Other studies performed in India have been successful in identifying CT using NAAT in similar populations, although prevalence was often low [32,33]. The serology results in this study appear to be slightly lower than other studies performed on OPD populations. It should be stated, however, that the studied population is rural and may hence be at a lower risk compared to urban settings, where most of the studies on *C. trachomatis* in India were performed. Higher *C. trachomatis* prevalence was in fact reported by Ghosh et al. in Kolkata on infertile women using both PCR and ELISA [34]. Higher rates of *C. trachomatis* were also report in New Delhi among symptomatic attendees of the gynecology OPD [35]. In another study from Nellore in the southern part of India, Vidwan et al. also pointed out an extremely low prevalence of *C. trachomatis* in a population of pregnant women [36].

Nonetheless, CT IgG was detected in the present study in 7.4% of the women. Since IgG positivity can be caused by both urogenital and ocular CT infection, we contacted the community ophthalmologist, who reported no cases of trachoma (CT serovars A–C) in the last ten years (personal communications). It hence seems likely that the ELISA based CT IgG positivity observed in this study reflects genito-urinary infections (CT serovars D–K) [6,37]. This means that the observed trends are most likely not due to an active CT infection in the lower genital tract at the moment of enrolment, but rather to an undetected upper genital tract infection, or to past infections (IgG) [38]. There is no reported evidence in the literature about the association between an irregular menstrual cycle and CT. It should, however, be stated that positive CT serology results may be caused by infections that took place in the past, which may deter the direct relationship between these two factors [39,40]. Some of the positive serology results may be imputable to upper genital tract infections. These are, however, more difficult to investigate as vaginal swabs would not test positive for CT DNA. Identifying a relation between an irregular menstrual cycle and an upper genital tract infection would necessitate a difficult study design, and one would expect to find some remnant CT participles shed into the vagina, while the current study does not [3,41,42,43]. Discussing symptoms with partners was also linked (*p* = 0.07) to CT positivity, but not significantly, and one should realize that CT infections in women are mostly (80–85%) asymptomatic [44]. Finally, the association with tenesmus (painful and ineffective inclination to evacuate the bowels), is not a common symptom of rectal CT infections, but some case reports have been described in women [45,46]. The study, however, did not test for CT presence in the rectal tract among the included women. Further research is needed to prove whether these associations are real and if the cohort needs to be extended. Such effort are currently ongoing, also to exclude linkage based on coincidence.

This paper analyzed also the characteristics and behavior of the women included in the study, and correlated it with serology outcomes for CT. Looking at such data, the low prevalence of *C. trachomatis* infection in the present population could be explained by the low risk profiles of these women. The great majority of the study participants are married housewives, which contributes to a lower likelihood for them to engage in promiscuous and high-risk behavior. The low-risk profile of most of the participants in the study may also explain the low use of protective measures, such as condoms.

The high proportion of women who presented with symptoms stresses the need to distinguish between physiological and pathological discharge in clinical settings [30]. A large proportion of the patients examined in this study in fact presented with vaginal discharge, although some positive cases of STIs could be identified. These findings are also interesting, as a significant proportion of infections with *C. trachomatis* and a lower proportion with *N. gonorrhoeae* often remain asymptomatic. In the present study, patients tested negative for some STIs, but did present with symptoms generally characteristic of them. The results could also be interpreted as biases in the testing. It has been reported and highlighted before that consumption of antibiotics, particularly broad spectrum, is common in the Indian communities [47]. Therefore, more research is necessary to investigate the proportion of physiological discharge, as compared to the discharge that arises from an infection with a sexually transmitted pathogen. Another important consideration is that the population was sampled out of a gynecology OPD, where complaints such as discharge may be common and associated with a wide variety of causative factors. This may have led to the overrepresentation of such symptoms in the studied population.

The questionnaire data point at a high burden of adverse pregnancy outcomes. Thirty-one percent of the study subjects had indeed experienced one or more miscarriages. The amount of recurrent miscarriages (instances of two or more) was also very high, with a rate of 12% (37/296). The rate in the present study stands above the rates put forward by an American systematic review from 2012, that identified a range of 11–22% for miscarriages [48], and by other studies in India [49]. Similarly, this study identifies high prevalence rates of stillbirths (24/296, 8%). These values stand above what is reported from both high-, as well as low- and middle-income countries [50]. The fact that the sampling for the present study took place within a healthcare setting that deals primarily with conditions such a miscarriages, stillbirths, and other complaints may cause those to be overrepresented in the current study population.

The main limitation of the present study lies in the sample size. It can be theorized that some of the correlations between certain factors and the CT seropositivity would have reached significance with a large sample size. A further limitation lies in the absence of additional testing methods for NG. Bacterial culture could in fact have helped to confirm the absence of NG cases. The testing facilities, however, lacked the capacities to perform NG culture. Furthermore, the negativity of CT results was confirmed by an in-house PCR assay that was performed alongside the HLA typing. This assay was performed on a subset of samples and targeted the chlamydial plasmid and the ompA gene. All of the tested samples came out as negative, while all of the positive controls were identified as positive. More research would be beneficial to investigate SRH awareness in India to identify knowledge gaps [51,52]. This could be addressed through culturally sensible education programs. Additional research in similar populations (women attending the OPD) and high-risk groups (men who have sex with men) have been initiated in the follow-up of this study in a different project site in southern India. These research endeavors capitalize on the conclusions of the present study and will also investigate OTC antibiotic use, as well as barriers and stigma that deter patients from seeking care. Preliminary results from the testing sites have already highlighted some CT and NG positivity.

## 5. Conclusions

The current study provides a snapshot of the SRH characteristics within a community of semi-urban women in the district of Allahabad, in North India. The findings indicate a situation with a wide range of sexual and reproductive health presentations, such as vaginal discharge and abdominal pain, and high rates of adverse pregnancy outcomes such as stillbirths and miscarriages. Although the burden of STIs was reported to be low in the tested population, symptoms such as pain and vaginal discharge were common. This raises the question of the importance and combinations of symptoms, and the differentiation between physiological and pathological discharge. New research insights would be necessary to better recognize complaints and manage them accordingly. New approaches to meet the SRH needs of women and communities across the Indian landscape should be put in place. New methods ought to focus on empowering women to seek care and reduce stigmatization. The data from this study also suggest that a non-negligible part of the study population faces sexual violence, and some individuals have very early sexual debuts. This touches upon a complex mosaic of factors that might contribute to the vulnerability of women within the Indian society. It is also important to stress the pressing need for education and increased awareness of the matters touched upon in this paper. Gaps in knowledge surrounding STIs and SRH hence ought to be identified and addressed in a way that is culturally appropriate and that can be readily taken up by the communities in India. Establishing a knowledge base is essential to improve upon the sexual and reproductive health situation in India.

## Figures and Tables

**Table 1 microorganisms-07-00557-t001:** Characteristics of the study population (*n* = 296).

Demographic Characteristics		Count (%)
Age (mean and range)		29 (18–72)
Marital status		
Married		290 (98%)
Widowed		3 (1%)
Divorced		3 (1%)
Number of children (mean and range)		1.5 (0–7)
Number of children (distribution)	0	85 (28.7%)
	1	79 (26.6%)
	2	72 (24.3%)
	≥3	60 (20.2%)
Number of pregnancies (mean and range)		2.5 (0–8)
Number of pregnancies (distribution)	0	22 (7.4%)
	1	79 (26.6%)
	2	78 (26.3%)
	≥3	117 (39.5%)
Ethnicity		
Indian/Hindu		210 (70.9%)
Indian/Muslim		66 (22.3%)
Indian/Christian		20 (6.7%)
Level of education		
Illiterate		39 (13.1%)
Less than primary		79 (26.7%)
Secondary		65 (21.9%)
Graduate		78 (26.3%)
Post-graduate		35 (11.8%)
Employment		
Housewife		253 (85.4%)
Teacher		10 (3.3%)
Professor		9 (3%)
Other		24 (8%)
*Chlamydia trachomatis* (CT) positive (IgG ELISA) *		22 (7.4%)
*Chlamydia trachomatis* positive (NAAT)		0 (0%)

(*) CT positivity was defined as CT IgG positive.

**Table 2 microorganisms-07-00557-t002:** Symptoms and sexual and reproductive health presentations in the testing population in relation to Chlamydia positivity (CT+) and negativity (CT−) (*n* = 296).

Symptoms and Presentations		CT+	CT−	Missing	*p*-Value
Vaginal discharge	Yes	21	254	0	1
	No	1	20		
Abdominal pain	Yes	17	186	0	0.47
	No	5	88		
Dysuria	Yes	9	114	0	1
	No	13	160		
Dyspareunia	Yes	2	31	0	1
	No	20	243		
Warts	Yes	10	100	0	0.49
	No	12	174		
Tenesmus	Yes	4	20	0	0.07
	No	18	254		
Menstrual pattern	Regular	17	246	9	0.02
	Irregular	5	19		

**Table 3 microorganisms-07-00557-t003:** Behavioral aspects in the studied population (*n* = 296).

Behavioral Factors		CT+	CT−	Missing	*p*-Value
Sexually active in last 6 months	Yes	13	260	1	0.66
	No	2	20		
Age of sexual debut	13–20	17	184	0	0.17
	21–33	5	90		
Consumption of alcohol	Yes	13	95	1	0.03
	No	9	178		
Use of male condoms	Yes	4	58	1	1
	No	18	215		
Experienced force during sex	Yes	11	129	1	0.82
	No	11	144		
Discussed complaints with partners	Yes	14	115	0	0.07
	No	8	159		

**Table 4 microorganisms-07-00557-t004:** Obstetrical and pregnancy history of the participants (*n* = 296).

Obstetrical History		CT+	CT−	Missing	*p*-Value
Miscarriages	0	18	184	0	0.91
	1	3	55		
	>1	1	35		
Number of pregnancies	0	2	20	0	0.86
	1	6	73		
	2	6	72		
	≥3	8	109		
Number of stillbirths	0	20	251	0	1
	≥1	2	23		

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
