# Peer review of "Sexually Transmitted Infections and Behavioral Determinants of Sexual and Reproductive Health in the Allahabad District (India) Based on Data from the ChlamIndia Study"

_microorganisms, 2019, doi:10.3390/microorganisms7110557_

Round 1

Reviewer 1 Report

The manuscript by Thomas et al reports the results of an STI epidemiological study performed in Uttar Pradesh India from 2016-2018. The overall data from this study are a meaningful contribution to the field, and highlight a need for additional scrutiny in developing regions such as the one examined. Of a minor note, the manuscript contained numerous spelling and typographical errors; some of these are indicated below, but I suggest the authors read the entire manuscript for accuracy.

Specific comments

Discussion runs long, perhaps more than is necessary for the scope of the study. The suggestion that excessive antibiotic use might be responsible for the negative NAAT results is intriguing, yet perhaps highly speculative. I am curious to know if the authors attempted to run alternative NAAT assay(s), even on a subset of the samples? Furthermore, were any attempts made to determine NG positivity by an alternative approach?

Minor comments

1. Typos on lines: 24, 59, 72, 94, 107, 108, 140, 156, 209, 225, 314
2. Line 133: Unclear what the * footnote refers to (no * in the table)
3. Line 138: subtitle not needed

Author Response

Dear Reviewer,

Many thanks for taking the time to review our manuscript and for providing valuable insights. We have taken your comments into account and made the following changes:

-The manuscript by Thomas et al reports the results of an STI epidemiological study performed in Uttar Pradesh India from 2016-2018. The overall data from this study are a meaningful contribution to the field, and highlight a need for additional scrutiny in developing regions such as the one examined.

à Thank you for this comment

-Of a minor note, the manuscript contained numerous spelling and typographical errors; some of these are indicated below, but I suggest the authors read the entire manuscript for accuracy.

à We have checked the manuscript for typos and grammatical mistakes

Specific comments:

-Discussion runs long, perhaps more than is necessary for the scope of the study.

à We have shortened the discussion by 500 words and removed the parts that were least in line with the scope of the paper.

-The suggestion that excessive antibiotic use might be responsible for the negative NAAT results is intriguing, yet perhaps highly speculative. I am curious to know if the authors attempted to run alternative NAAT assay(s), even on a subset of the samples?

à We agree that concluding that the consumption of OTC antibiotics among the study participants may appear speculative at first. However, this hypothesis was put forward by the team of physicians (gynecologists and obstetricians) that have worked closely with the community and testing populations for a while. Furthermore, these results came as a surprise for the whole testing team, we therefore worked out the different causes which could have led to CT and NG negativity. The PRESTO PCR assay has been used in different settings in both developed and developing countries and was successful in identifying prevalence below 10%. We have added references to these studies at line 201 in the discussion. Furthermore, the samples collected for this study were processed by a trained Dutch research and expert in utilizing the PRESTO kit and members of the local research team. It is hence unlikely that errors in manipulation of the samples could have caused the testing to turn out unsuccessful. Furthermore, the PRESTO kits feature positive internal controls, which all tested positive in the study, indicating that the negativity could not be explain by shortcomings in the amplification procedure. All in all, in spite of its speculative nature, the hypothesis that the PCR results remained the most plausible one. We have furthermore initiated new studies in a different location in India and the preliminary result indicate positive results on the PCR platform using the PRESTO testing kit. You can consult on these results from line 357 to 364 in the discussion.

Furthermore, we have added some supplementary information on the additional testing that was performed on a subset of samples using an in-house PCR assay. This assay was performed alongside the HLA typing and targeted the chlamydial plasmid and the OPM-1 gene. This assay also failed to highlight any CT positivity.

Furthermore, were any attempts made to determine NG positivity by an alternative approach?

à We also have also added some reflection on the absence of additional testing for NG. On the one hand, Serology testing for NG is not routinely performed as there are no reliable tests available. One option could have been to culture potential NG samples. The testing facilities at the project site were however not capable to perform culture. Please feel find these changes in the discussion from line 340 to line 347.

Minor comments:

Typos on lines: 24, 59, 72, 94, 107, 108, 140, 156, 209, 225, 314 Line 133: Unclear what the * footnote refers to (no * in the table) Line 138: subtitle not needed

à We have corrected all of the typos that you highlighted and screened the paper for further typos and corrected them

Reviewer 2 Report

Clearly a well done and important study, however the findings being somewhat inconclusive is related to both the size of the sample and possibly the type of population chosen. Would suggest future studies be directed at populations with different risk profiles such as clientele of STI clinics as well as services directed towards HRG such as MSM; FSW etc.

Author Response

Dear reviewer

Many thanks for taking the time to review our manuscript and for providing valuable insights. We have taken your comments into account and made the following changes:

Reviewer 2 Clearly a well done and important study, however the findings being somewhat inconclusive is related to both the size of the sample and possibly the type of population chosen. Would suggest future studies be directed at populations with different risk profiles such as clientele of STI clinics as well as services directed towards HRG such as MSM; FSW etc.

à We have added an overview the studies and research projects that follow the present study. We have outlined the fact that we learned from the present study, and we have included insights from the preliminary findings. Please refer to lines 357 to 364 to find these changes in the paper.

Round 2

Reviewer 1 Report

I recommend that the authors read the entire manuscript carefully for typographical errors. Beyond that, only minor items are suggested for editing:

In the clean version:

1. Line 213

2. Line 296: should be ompA not OMP-1

Author Response

Dear Reviewer,

Thanks again for taking time to go through our manuscript. We have corrected the points that you pointed out and performed a thorough check of the paper that led to many corrections and improvements besides the ones that you brought to our attention. 

We hope to have addressed your comments sufficiently and appropriately. 

Many thanks for supporting our research 

Pierre Thomas, also on behalf of the ChlamIndia research team. 

This manuscript is a resubmission of an earlier submission. The following is a list of the peer review reports and author responses from that submission.